# Exploiting Comparative Omics to Understand the Pathogenic and Virulence-Associated Protease: Anti-Protease Relationships in the Zoonotic Parasites *Fasciola hepatica* and *Fasciola gigantica*

**DOI:** 10.3390/genes13101854

**Published:** 2022-10-14

**Authors:** Krystyna Cwiklinski, John Pius Dalton

**Affiliations:** 1Institute of Infection, Veterinary and Ecological Sciences, University of Liverpool, Liverpool L69 3BX, UK; 2Molecular Parasitology Laboratory (MPL), Centre for One Health and Ryan Institute, School of Natural Sciences, University of Galway, H91 DK59 Galway, Ireland

**Keywords:** *Fasciola*, flukes, trematodes, worms, helminths, genomics, transcriptomics, proteomics, peptidases, peptidase inhibitors

## Abstract

The helminth parasites, *Fasciola hepatica* and *Fasciola gigantica*, are the causative agents of fasciolosis, a global and economically important disease of people and their livestock. Proteases are pivotal to an array of biological processes related to parasitism (development, feeding, immune evasion, virulence) and therefore their action requires strict regulation by parasite anti-proteases (protease inhibitors). By interrogating the current publicly available *Fasciola* spp. large sequencing datasets, including several genome assemblies and life cycle stage-specific transcriptome and proteome datasets, we reveal the complex profile and structure of proteases and anti-proteases families operating at various stages of the parasite’s life cycle. Moreover, we have discovered distinct profiles of peptidases and their cognate inhibitors expressed by the parasite stages in the intermediate snail host, reflecting the different environmental niches in which they move, develop and extract nutrients. Comparative genomics revealed a similar cohort of peptidase inhibitors in *F. hepatica* and *F. gigantica* but a surprisingly reduced number of cathepsin peptidases genes in the *F. gigantica* genome assemblies. Chromosomal location of the *F. gigantica* genes provides new insights into the evolution of these gene families, and critical data for the future analysis and interrogation of *Fasciola* spp. hybrids spreading throughout the Asian and African continents.

## 1. Introduction

Liver fluke parasites of the genus *Fasciola*, such as *F. hepatica* and *F. gigantica*, undergo a complex life cycle involving a snail intermediate host and mammalian definitive host (Figure 1; [1]). Within both hosts, the liver flukes undergo rapid morphogenesis into distinct developmental stages that are confronted with different macromolecules, microenvironments, tissues, and cells. The *Fasciola* spp. parasites have adapted to infect a broad range of mammals and several snail species, which accounts for their extensive geographical distribution and prevalence across the globe [2,3]. Despite having a wide host range, the mechanism by which these parasites invade the host tissues and migrate to their next tissue destination is universal, and is facilitated by the molecules they release at the parasite-host interface.

Proteomic studies of the molecules released by the *F. hepatica* stages in the mammalian host, termed excreted-secreted (ES) products, found that a large proportion of these host interacting proteins were comprised of highly proteolytic cathepsin cysteine peptidases (reviewed by [4]). Further studies revealed that the activity of these peptidases is strictly regulated by the co-release of a variety of cognate peptidase inhibitors, namely Kunitz-type inhibitors, and stefins/cystatins [5,6,7]. In addition to regulating the parasite cathepsin peptidases, we have shown that the peptidase inhibitors also play an important role in regulating host lysosomal-like cathepsin peptidases involved in the immune response to parasite infection [5,6,7]. Besides cysteine peptidase inhibitors, we discovered that *F. hepatica* also secretes a range of serine peptidase inhibitors (serpins), which have no parasite peptidase target but are exclusively applied to block host serine peptidases, for example Mannose Binding Serine Proteases (MASP) that are critical to complement activation via the lectin pathway [8,9].

Genes encoding the cathepsin peptidases and the inhibitors of serine and cysteine peptidases belong to multi-membered gene families, thought to have evolved by gene duplication followed by structural/functional diversification [4,5,7,9,10,11]. Deciphering the number and structure of genes contributing to these peptidase families is reliant on robust genome assemblies. Unfortunately, except for the recent chromosomal level *F. gigantica* genome, the remaining *F. hepatica* and *F. gigantica* genome assemblies are comprised of thousands of scaffolds [12,13,14,15]. Furthermore, because the members of these gene families tend to share high levels of sequence identity, their identification and phylogenetic relationships can be difficult to tease apart. Therefore, constant refinement of these gene family models is required as the genomic/transcriptomic data continues to evolve.

The tropical parasite *F. gigantica* is often less well studied and researched compared to its temperate counterpart, *F. hepatica*. Although several *F. gigantica* cathepsin peptidases and peptidase inhibitors have been reported recently [16,17,18,19,20,21,22,23,24], no extensive study of these gene families has been carried out to date. Comparative gene analysis has shown that approximately 70% of genes are shared between *F. hepatica* and *F. gigantica* [25,26]; however, parallel analysis of comparable life cycle stages suggests that the transcriptional profile of these genes exhibits species-specificity [26]. These observations predict that although these closely related parasites, which diverged about 5 million years ago [14], utilise homologous peptidases to achieve their in vivo goals, there may be subtle differences in the way they are used at various developmental stages in response to the specific environmental parameters or cues that each parasite encounters. 

The availability of several *F. hepatica* and *F. gigantica* genome assemblies, together with life cycle stage-specific transcriptome datasets, heralds a new era in our ability to genetically interrogate and compare these two zoonotic parasites of major global importance. Accordingly, given our interest in parasite peptidases and their role in host–parasite interactions, we have exploited this new data to refine our identification of the peptidase and peptidase inhibitor families within these datasets and show how new understandings can be uncovered by comparing the genomes of these two parasites. Since peptidases play an important role in liver fluke biology, pathogenesis and virulence, this data will be used to assist future diagnostics and vaccine development.

## 2. Materials and Methods

### 2.1. Fasciola spp. Databases Used to Isolate/Identify Peptidase and Peptidase Inhibitor Genes

In this study we have re-analysed our previously published *F. hepatica* and *F. gigantica* omics datasets to identify the peptidase and peptidase inhibitor genes [12,26,27,28,29] and carried out comparative analyses with the recently available *F. hepatica* egg transcriptome and proteome data published by Ilgová et al. [30] and *F. gigantica* genome assemblies [14,15] as detailed below.

Analysis by BLASTp against the MEROPS collection (release 11.0; www.ebi.ac.uk/merops, accessed on 30 August 2022; default settings, accessed 21 July 2022 [31]) followed by manual annotation was carried out to determine the peptidase and peptidase inhibitor gene family profile using the following *Fasciola* spp. datasets: (1) *F. hepatica* analysis was based on the gene models identified within the *F. hepatica* genome (PRJEB6687; PRJEB25283; [12]) and by parsing the data from the study of the *F. hepatica* egg transcriptome (GSE160622) published by Ilgová et al. [30]; (2) *F. gigantica* stage-specific transcriptomes (PRJNA350370) reported by Zhang et al. [26]. The transcript and protein expression profiles were extracted from the stage-specific transcriptome and proteome datasets for *F. hepatica* [12,27,28,29] and *F. gigantica* [26]. The resulting data was graphically represented by ggplot2 in R.

### 2.2. Identification of the Gene Families Relating to the Cathepsin Peptidases and Peptidase Inhibitors

Previously characterised *F. hepatica* sequences identified as cathepsin L peptidases, cathepsin B peptidases, legumain, Kunitz-type inhibitors, serine protease inhibitors (serpins) and stefins/cystatins from the following studies [4,5,7,9,12] were used as reference sequences for BLASTp analysis to (a) confirm the sequences within the *F. hepatica* genome, and (b) identify homologous sequences within the stage-specific *F. gigantica* transcriptome datasets (PRJNA350370). The *in silico* descriptive annotations of the gene transcripts in the specific datasets were also screened. All the sequences were manually assessed, and their annotation and putative domains identified by InterPro analysis (www.ebi.ac.uk/interpro, accessed on 30 August 2022). The transcript and protein expression profiles were extracted from the stage-specific transcriptome and proteome datasets for *F. hepatica* [12,27,28,29] and *F. gigantica* [26]. The transcriptome data was graphically represented using heatmaps generated using pheatmap in R.

### 2.3. F. gigantica Genome Analysis

The *F. gigantica* cathepsin peptidase and peptidase inhibitor sequences identified within the stage-specific transcriptomes were confirmed/assessed using BLASTn and BLASTp against the *F. gigantica* genome assemblies (PRJNA230515, [14]; GWHAZTT00000000, [15]). Determination of the chromosomal position of the genes was based on identifying the homologous sequence in the chromosomal level *F. gigantica* genome sequence and extrapolating the data from the study by Luo et al. [15].

## 3. Results

### 3.1. Peptidases and Peptidase Inhibitors Expressed throughout the Life Cycle of Fasciola spp.

The publicly available *Fasciola* spp. genome and transcriptome datasets were interrogated to identify genes encoding peptidase and peptidase inhibitors, based on MEROPS classification [31], that are expressed during the complete parasite life cycle (Figure 2; Appendix A). Consistent with previous studies of *F. hepatica*, the class of peptidases predominantly expressed by the stages associated with infection in the mammalian host, namely metacercariae, NEJ, immature and adult flukes (Figure 1), are cathepsin-like cysteine peptidases. In contrast, peptidase genes associated with a variety of catalytic activities are transcribed in the embryonating eggs, with a particular majority encoding aspartic peptidases. This wider profile of peptidases is also observed within the intra-snail stages, the miracidia and rediae, with comparable levels to that expressed by the eggs. However, once the parasites have developed onto the cercarial stage that emerges from the snail, they display a similar and more restricted profile like the next stage, the infective metacerceriae.

Notably, the profile of peptidase inhibitors is more dynamic than their cognate peptidases, with a range of serine, cysteine and metallopeptidase inhibitors being differentially transcribed throughout the life cycle. In contrast to the distinct pattern of transcription of the peptidases exhibited at each distinct life cycle stage, the inhibitors display different profiles. Moreover, these profiles are also not the same in the *F. hepatica* and *F. gigantica* datasets; *F. hepatica* most abundantly transcribes serine peptidase inhibitors (I04) across its life cycle stages, whereas Kunitz-type inhibitors (I02) predominate in *F. gigantica*. Our previous studies have shown that the *F. hepatica* Kunitz-type inhibitors are unique because they potently inhibit cathepsin cysteine peptidases in addition to trypsin-like serine proteases, rather than being exclusive serine peptidase inhibitors [5,6]. Nevertheless, it is clear that both parasite species invest considerable energy in transcribing genes to produce proteins capable of inhibiting cysteine peptidases, consistent with the abundant expression of this proteolytic enzyme type.

Another key difference observed from comparing the *F. hepatica* and *F. gigantica* datasets is the biased abundance of inhibitors of the class I39 (mammalian alpha2-macroglobulin and other large homologous proteins that interact with endopeptidases regardless of catalytic type) in *F. hepatica* adult fluke. The function of these inhibitors is unclear but differences between the mammalian hosts from which the adult *F. hepatica* and *F. gigantica* were recovered would be an obvious starting point to examine their impact on the types of inhibitors that each parasite transcribes.

Temporal analysis of the *F. hepatica* somatic proteome reveals that high levels of cysteine peptidases are expressed across multiple life cycle stages, including the eggs, alongside an abundance of serine and cysteine peptidase inhibitors (I02, I04, I25; Appendix A). Despite observing lower transcriptional levels of metallopeptidase inhibitors (I63) in *F. hepatica* eggs compared to *F. gigantica*, protein products of these genes are abundant within the *F. hepatica* egg somatic proteome. It is worthwhile noting, however, as highlighted in the study by Ilgová et al. [30], that *Fasciola* eggs are laid unembryonated and as the eggs develop, their transcriptional and protein profiles change.

NEJ, immature and adult parasites secrete in vitro a similar profile of cysteine peptidases and serine and cysteine peptidase inhibitors, in high abundance (Appendix A; Appendix A). The majority of the cysteine peptidases belong to the papain-like cathepsin peptidase group (C01), with representation of the asparaginyl endopeptidases (legumain; C13) and peptidase family C56 (4-methyl-5(B-hydroxyethyl)-thiazole monophosphate biosynthesis protein). Metallopeptidases, specifically leucine aminopeptidases (M17) and dipeptidase/dipeptidylpeptidases (M24/M49), are more abundantly secreted by the adult stage parasites.

### 3.2. F. hepatica Cathepsin Cysteine Peptidases

In contrast to their mammalian hosts that express a wide range of cysteine peptidases (11 functional papain-like cysteine peptidases; [32]), only cathepsin L and cathepsin B cysteine peptidases are expressed by the *Fasciola* spp., as observed by Tort et al. [33], both of which are encoded by large gene families. Based on the current *F. hepatica* genome assemblies, which are comprised of several thousand scaffolds, at least 23 cathepsin L peptidase genes and 15 cathepsin B peptidase genes have been identified, represented by both partial and complete sequences [4]. Eight legumain genes encoding asparaginyl endopeptidase enzymes that regulate the activation of cathepsin peptidases [34], were also identified.

Differential gene transcription analysis highlights that these genes are strictly stage and host-specific, with separate genes being expressed by the stages associated with the snail and mammalian hosts, respectively (Figure 3; Appendix A; [12,26]). This highlights another level of peptidase activity regulation employed by these flatworm parasites that is intricately tied to their lifecycle and development.

### 3.3. F. gigantica Cathepsin Cysteine Peptidases

Comparative analysis of the *F. gigantica* cathepsin peptidase and legumain genes within the stage-specific transcriptome datasets uncovered 11 cathepsin B genes that share sequence identity and transcriptional profiles with *F. hepatica* (Figure 3; Appendix A). In contrast, not all homologs of the *F. hepatica* cathepsin L peptidase and legumain genes could be identified in the *F. gigantica* transcriptome datasets. This result was confirmed by analysis of the *F. gigantica* genome described in Section 3.5 below.

Twenty-five *F. gigantica* cathepsin L gene transcripts were identified (Figure 3; Appendix A). In *F. hepatica* the cathepsin L3 clade expressed predominantly in the juvenile mammalian stages is thought to be the ancestral cathepsin L gene from which the remaining cathepsin L genes duplicated and diverged; the *F. hepatica* cathepsin L peptidases separate into five clades, of which the cathepsin L3 clade is comprised of four genes [4,10]. Consistent with the critical role of the cathepsin L3 peptidases in tissue degradation and host invasion since they exhibit unique collagenase-like activity [35,36,37], *F. gigantica* expresses three cathepsin L3 genes. Genes representing clades one, two and five were also identified (*Fg_CL1_2/4* & *FgCL5*); however, we could not differentiate one gene between clade one and two based on current sequence analysis (*FgCL1_6/CL2*). The remaining 19 cathepsin L gene transcripts matched four of the partial sequences described by Cwiklinski et al. [4] (*FgCL_2, FgCL_3, FgCL_4, FgCL_5*) and other cathepsin L-like genes not previously described, which were predominately transcribed by the intra-snail developmental stages. The most abundantly transcribed cathepsin L genes corresponded to *FgCL_4* and *FgCL_5* expression by the rediae, *FgCL3_3* and *FgCL3_4* expression by the cercariae and metacercariae stages, and *FgCL1/2* and *FgCL5* expression by the immature and adult fluke.

Fifteen legumain gene transcripts were identified in the *F. gigantica* datasets (Figure 3; Appendix A), corresponding to legumain 1 (*FgLeg1*), legumain 2 (*FgLeg2*), legumain 3 (*FgLeg3*), legumain 4 (*FgLeg4*) in addition to 11 legumain-like genes that require further characterisation. The most abundant transcription was observed for *FgLeg1* in the cercariae and metacercariae stages, *FgLeg-like 10* and *FgLeg-like 11* in the immature flukes, and *FgLeg3* and *FhLeg4* in both immature and adult fluke.

Cathepsin L peptidases are first synthesized as inactive precursors, termed pro-enzymes or zymogens, which become activated in the low pH of the parasite gut lumen by removal of a N-terminal extension or propeptide [38]. Removal of the propeptide can occur by either (a) autocatalytic intra-molecular cleavage whereby a molecule of active cathepsin L removes the propeptide of another cathepsin L molecule, (b) trans-molecular cleavage of the cathepsin L propeptide by a co-secreted legumain/asparaginyl endopeptidase, or (c) both intra- and trans-molecular cleavage happening together. We have previously shown that intra- and trans-molecular cleavage points occur at the junction between the propeptide and mature enzyme domain and that cleavage sites are highly conserved amongst the members of the *F. hepatica* cathepsin L family. Here, we found that both cleavage sites are also conserved amongst the cathepsin L peptidases of *F. gigantica* (Figure 4).

### 3.4. Key Fasciola spp. Peptidase Inhibitors

Highlighted by the MEROPS analysis, the *Fasciola* spp. parasites transcribe a dynamic range of peptidase inhibitors that are mainly focused on the inhibition of cathepsin cysteine peptidases and serine peptidases. Key inhibitors involved in these processes are the Kunitz-type cysteine/trypsin protease inhibitors, broad-range serine protease inhibitors (serpins) and cysteine peptidase inhibitors (stefins/cystatins) that we have previously shown are encoded by multi-membered gene families [5,7,9].

Our current deeper analysis of the available *F. hepatica* genomic and transcriptomic data (Figure 5; Appendix A) identified sequences corresponding to 11 single domain Kunitz-type inhibitors, including the seven genes previously described by Smith et al. [5]. A further four multi-domain Kunitz-type sequences were also identified that possess up to ten Kunitz-like domains and that share similarity with papillin-like and spondin-like proteins. No additional *F. hepatica* sequences related to the stefin/cystatin-type (3 members) and serpin-type inhibitors (7 members) were discovered in this study.

Comparative transcriptomic analysis clarified that *F. gigantica* expresses several homologous peptidase inhibitor sequences to *F. hepatica* (Figure 5; Appendix A). Three stefin genes corresponding to Stefin1 (*FgStf1*), Stefin 2 (*FgStf2*) and Stefin3 (*FgStf3*) were identified, in addition to the multi-domain cystatin (*FgCys1*), consistent with previous reports from our and other laboratories [7,11,16]. Sequences corresponding to the five *F. hepatica* phylogenetic Kunitz-type inhibitor groups (nomenclature from [5]) were identified, in addition to two single domain Kunitz genes (*FgKT_A* and *FgKT_B*) and five multi-domain Kunitz genes (*FgKT_E*–*FgKT_I*). Only four serpin sequences were identified within the *F. gigantica* transcriptome datasets that share sequence identity with *FhSrp1/FhSrp3*, *FhSrp5*, *FhSrp6*, *FhSrp7* (nomenclature from [9]).

We found that the peptidase inhibitor genes undergo tightly controlled temporal expression throughout the *Fasciola* spp. life cycle. Where life cycle stages between *F. hepatica* and *F. gigantica* could be compared, similar expression levels of transcription were observed between the homologous genes. However, with respect to the multi-domain Kunitz genes we observed that these were predominately transcribed by the *F. hepatica* stages associated with the mammalian host whereas in *F. gigantica* these genes are mainly expressed by the miracidia and metacercariae life cycle stages.

Consistent with the MEROPS analysis, the most abundantly transcribed *F. gigantica* peptidase inhibitor is the Kunitz-type inhibitor, *FgKT1*, produced and secreted by the immature fluke stages. While the three stefin genes are transcribed at high levels throughout the lifecycle, their highest transcriptional levels are observed within the cercariae and metacercariae stages for *FgStf1* and the immature flukes for *FgStf2* and *FgStf3*. The most transcribed serpin genes are *FgSrp1/FgSrp3* produced by the cercariae and metacercariae and *FgSrp6* by the eggs and miracidia.

### 3.5. Chromosomal Location of Key F. gigantica Cathepsin Peptidase and Peptidase Inhibitor Genes

To determine the specific number of genes that matched to the gene transcripts of cathepsin peptidases and the key peptidase inhibitors expressed by the *Fasciola* spp. parasites (Kunitz-type inhibitors, serpins and stefins/cystatins) we carried out comparative analyses with the available *F. gigantica* genome/chromosomal data. As expected, several of the transcript clusters identified within the *F. gigantica* transcriptome data mapped to only one position/gene, reducing the number of genes relating to these peptidases and peptidase inhibitors (Appendix A). Based on the data from the study by Luo et al. [15], the chromosomal location of these genes could also be determined (Figure 6; Appendix A).

#### 3.5.1. Peptidase Gene Families

Only nine genes corresponding to cathepsin L peptidases were identified in the chromosomal level *F. gigantica* genome assembly, and these were located on six of the ten chromosomes (chromosomes 1, 2, 3, 6, 7, 8). Most of these genes matched the cathepsin L genes described here as being abundantly transcribed by the intra-snail stages (Appendix A). The genes corresponding to the phylogenetic clades of *F. hepatica* cathepsin L genes mapped to four genes located on chromosome 6 and 7 (Appendix A).

This analysis also identified 12 cathepsin B genes located on three chromosomes (chromosomes 1, 4, 8), with the majority of genes located on chromosome 4 including the closely related *FgCB1*, *FgCB2* and *FgCB3* genes and multiple genes that could be annotated as *FgCB4*.

Eight legumain genes were identified, all located on chromosome three. The reduced number of legumain genes compared with the transcriptome analysis is due to several legumain-like genes mapping to the same region.

Corresponding analysis of the cathepsin L and B genes within the *F. gigantica* genome with the WormBase ParaSite database (PRJNA230515) identified 34 cathepsin genes. The majority of these genes could be annotated as cathepsin B genes, confirming the reduced cohort of cathepsin L-genes. However, mapping the chromosomal location of these genes identified several sequences within the genomic sequence that had not been designated as genes, indicating that significant refinement and characterisation of the cathepsin L and B gene families in *F. gigantica* is still required.

#### 3.5.2. Peptidase Inhibitor Gene Families

The *F. gigantica* single domain stefins (*FgStf1*, *FgStf2*, *FgStf3*) mapped to eight genes all located on chromosome three, whereas the multi-domain cystatin located to chromosome one (Appendix A). The serpin genes corresponding to *FgSrp 1/3*, *FhSrp2*, *FhSrp5*, *FhSrp6* and *FhSrp7* localised to chromosome one and are represented by nine genes. The single domain Kunitz-type inhibitors are all located on chromosome one with the exception of *FgKT4* which is located on chromosome two. Finally, the multi-Kunitz domain inhibitors were found on three chromosomes (chromosomes 1, 2, 8).

## 4. Discussion

Notwithstanding similarities in their life cycles, parasitic worms of the *Fasciola* spp. have diverged, evolved and adapted to particular environmental and biological niches for over 5 million years [14]. *F. gigantica*, referred to as the tropical liver fluke due to its wide distribution throughout Asia, Africa and the Middle East [39], is more tolerant of higher temperatures and exposure to direct sunlight compared to the temperate liver fluke *F. hepatica*, which is found in cooler climes (reviewed by [3]). The inclination for higher temperatures has also had a direct influence on *F. gigantica* development; for example, egg embryonation occurs more rapidly [40] and the *F. gigantica* intra-snail stages develop at higher temperature thresholds (16 °C compared with 10 °C for *F. hepatica* development; [41,42]), which allows multiple rounds of clonal expansion resulting in five redial generations rather than that observed for *F. hepatica* where a maximum of four generations have been reported [41,43,44].

Keys to furthering our understanding of how such adaptations have evolved are contained in the emerging range of sequencing datasets for the *Fasciola* spp. that provide the information for species-specific interrogation of liver fluke biology at a molecular level. Advances in sequencing technologies and the continual reduction in their costs has facilitated the re-sequencing of several *F. hepatica* isolates [45] and provide a comprehensive platform for future studies. We can now perform genetic investigations to determine what role environmental factors such as climate, temperature, species of the mammalian or snail host, or genetic parameters within *F. hepatica* isolates play in their global prevalence, distribution and host virulence.

Because of our general interest in peptidases in host–parasite interaction, in the present study we probed the available sequencing data to specifically highlight the changes the parasites undergo in relation to the profile of proteases and protease-inhibitors they utilise during their migration in their hosts. We have previously shown that peptidases are critical to many parasite-related functions including host invasion, tissue penetration, feeding and virulence [33]. Here, we confirmed that *Fasciola* spp. parasites rely on a predominance of cathepsin cysteine peptidases to perform these functions. Interestingly, we observed a reduced cohort of cathepsin L peptidases within the *F. gigantica* genome in comparison to *F. hepatica*, which was confirmed by probing the two available *F. gigantica* genome assemblies. Further analysis is now required to determine the exact number of cathepsin L and B peptidase genes, to discern their individual biological function and to explain why the tropical flukes require less biochemically diverse peptidases.

Our in silico analysis of the residues spanning the propeptide-mature domain junction of cathepsin L peptidases that are involved in auto-catalytic processing and transactivation by legumains, indicates a common mode of peptidase activation. In the 3-D structure of the cathepsin L peptidases this sequence is exposed and flexible and is susceptible to proteolytic attack [10,37]. Nevertheless, we have shown for *F. hepatica* that specific sites for intra- and trans-molecular cleavage by cathepsin Ls and legumains, respectively, within this junction are conserved (Figure 4). It is not surprising that we found here that *F. gigantica* retains this conservation in the cathepsin Ls as we have also observed it in more distant worm parasites, such as *Schistosoma mansoni* reviewed by [4,33]. However, it highlights the importance of the legumain family in peptidase control as their expression is also tightly regulated. For this reason, we have suggested that legumains represent a promising target for either novel drug- or vaccine-mediated interventions [34].

In agreement with the study by Ilgová et al. [30], we observed that the eggs express a more dynamic range of peptidases and peptidase inhibitors, and this is also exhibited by the miracidia and rediae. This analysis is also consistent with the transcriptome analysis by Zhang et al. [26] that described an upregulation of peptidase inhibitors in the egg transcriptome, and an enrichment of zinc ion binding and metallo-endopeptidase activity. The abundant expression of metallo-peptidase inhibitors (I63) also highlights the importance of this catalytic type of peptidase for these egg stages and the strict regulation the parasite imposes on them. Similarly, an abundance of threonine peptidases associated with the proteasome is observed in this intra-snail stage, reflective of the increased gene transcription by the rediae that facilitates their clonal expansion through multiple generations [26]. Clearly, the growth and morphogenesis that take place during egg embyonation, emergence of free-living miracidia, adaptation to the snail invasion and redial development require rapid tissue degradation and re-modelling in which peptidases play a critical role.

In addition to the C01 and C13 class of cysteine peptidases, adult *F. hepatica* flukes secrete an abundance of 4-methyl-5(B-hydroxyethyl)-thiazole monophosphate biosynthesis protein (C56) that is an important part of energy metabolism via its involvement in thiamine metabolism. In plants, this C56 type protein is a target for the anti-oxidant thioredoxin during oxidative stress [46,47]. Recently, we suggested that the *F. hepatica* thioredoxin (FhTrx), which is abundant in *F. hepatica* ES, may function outside the thiol-dependent antioxidant cascade in immunomodulation. Another function may be related to the co-secretion of C56 class cysteine peptidases, although this is a molecular interaction that needs further exploration [48].

We found that *F. hepatica* and *F. gigantica* express a similar array of peptidase inhibitors that is dominated by the cathepsin peptidase inhibitors, namely Kunitz-type inhibitors and cystatin/stefins, and by the serine peptidase inhibitors, termed serpins. Biochemical analyses have only been carried out for the *F. gigantica* cystatins/stefins, that exhibit inhibitory activity against a range of host and parasite cathepsin L and B proteases [16,20,21]. Comparative analyses of the *F. hepatica* cystatins/stefins indicate that they are preferential inhibitors of cathepsin L peptidases [11]; Dalton, personal communication], implying potential species-specific roles for these inhibitors.

In this study we used the available chromosomal level *F. gigantica* genome data to investigate the location of the peptidase and peptidase inhibitor genes, which were distributed across seven of the ten chromosomes. The genes comprising the single domain stefin and legumain families were located within close proximity, respectively, on chromosome three. The members of the other peptidase and peptidase inhibitor gene families were spread across multiple chromosomes, and those genes on chromosome one did not exhibit grouping according to their gene family type. Chromosomal-level *F. hepatica* genome assemblies are anticipated and will allow comparative analyses of the genomic location of these genes in both *Fasciola* spp., and will greatly inform our understanding of gene transcription in these worms and helminths generally by elucidating the properties of promoter regions and upstream enhancers.

In areas where *F. hepatica* and *F. gigantica* overlap, such as in Southeast Asia, China, Korea, and areas of the Middle East and Africa, species hybridisation has been observed [49,50,51,52,53,54]. To date the genetic characterisation of these *Fasciola* spp. hybrids has been restricted to the DNA analyses of the mitochondrial genes, *cox 1* and *nad1*, and the nuclear genes, *pepck* and *pold*, used for diagnosis of these intermediate forms (reviewed by [55]). In this focused study, we have identified distinctions between *F. hepatica* and *F. gigantica* relating to the array of cathepsin L peptidase genes, gene family structure, and the relative expression profile of peptidase and peptidase inhibitors at stages of the parasite’s life cycle. How these diversities contribute to the parasite growth, development and host relationship remains unclear; however, they could have major impacts on the relative pathogenicity and virulence of each parasite that are important not only to further our understanding of the biology of these parasites but also to identify traits that each confers on the biology of hybrid *Fasciola* spp. parasites.

## 5. Conclusions

The advancement of omics technologies has led to a major leap in our molecular understanding of liver fluke biology and will facilitate a truly multidisciplinary approach to investigating these parasites and the disease they cause. Ultimately, it is imperative that these studies lead to new ideas on host–parasite interactions that can be translated into robust experimental validation studies. New diagnostics, particular animal-side rapid tests, are badly needed to help farmers monitor infections on farm, to perform widespread surveillance studies, to accurately detect human infections, and also to distinguish *F. hepatica* from *F. gigantica* infections (as well as their hybrids). Omics databases now provide a wealth of information for us interrogate and investigate to identify the much-needed molecular vaccines that will move us away from environmentally damaging chemical treatments. Most importantly, this data is freely available to all researchers, with different research interests using different approaches to reach that vaccine goal. Here, we show how we can exploit this data using in silico tools to further understand molecules that we have had a particular interest in for many years, peptidases and their inhibitors, so similar studies can be made by other laboratories.

## Figures and Tables

**Figure 1 genes-13-01854-f001:**
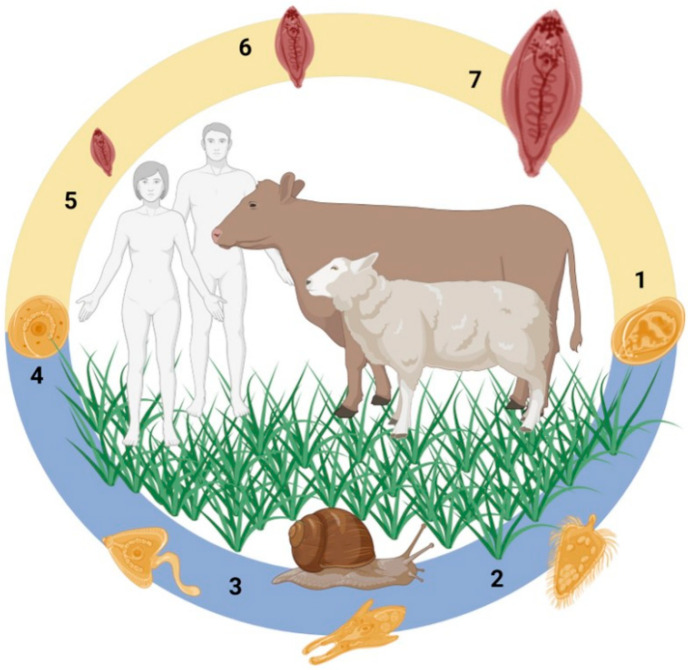
*Fasciola* spp. life cycle. (1) Eggs are passed in the faeces and undergo embryonation following appropriate temperature and moisture levels. (2) Following embryonation, the miracidia hatch from the eggs and search out the snail intermediate host. (3) Within the snail, the parasites undergo clonal expansion, developing through the rediae, sporocysts and cercariae stages. (4) The cercariae emerge from the snail following light and temperature cues and encyst as metacercariae that are observed on vegetation and floating in water. The metacercariae are the infectious stage that are ingested by the mammalian definitive host. (5) Following ingestion, the metacercariae excyst in the duodenum as Newly Excysted Juveniles (NEJ) that migrate across the gut wall via the peritoneal cavity to the liver. (6–7) Once in the liver, the immature flukes rapidly grow and develop while migrating through the liver parenchyma to the bile ducts, where the mature adult parasites reside, releasing thousands of eggs per day. Figure created using Biorender; Parasite medical art provided by Les Laboratories Servier, https://smart.servier.com, accessed on 30 August 2022.

**Figure 2 genes-13-01854-f002:**
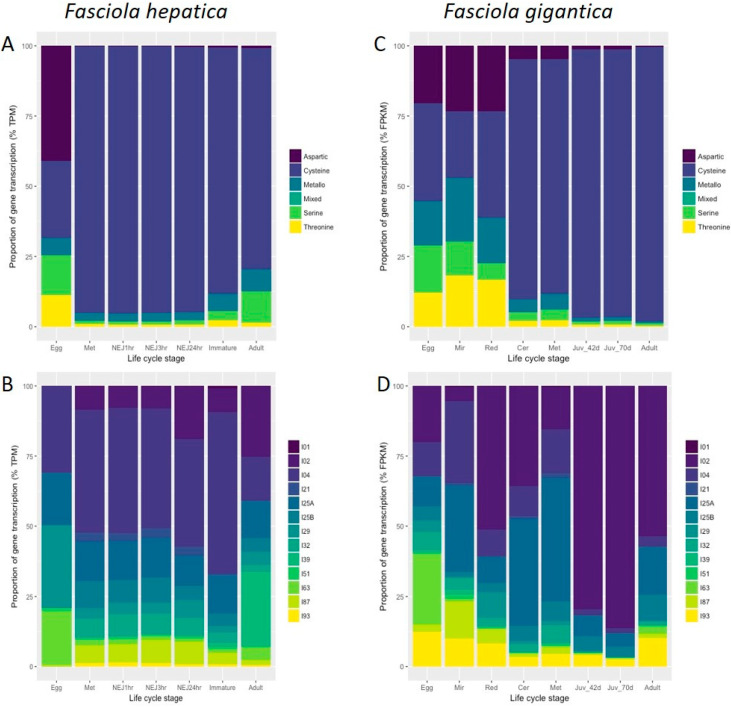
Graphical representation of the proportion of gene transcription relating to peptidase and peptidase inhibitor families throughout the *Fasciola* spp. life cycle. (**A**,**B**) The transcriptional profile of *F. hepatica* based on the percentage transcripts per million (TPM) of the respective peptidase (**A**) and peptidase inhibitors families (**B**) from the stage-specific transcriptomes described by [12,30]. (**C**,**D**) The transcriptional profile of *F. gigantica* based on the percentage fragments per kilobase of exon per million mapped fragments (FPKM) of the respective peptidase (**C**) and peptidase inhibitors families (**D**) from the stage-specific transcriptomes described by [26]. Peptidase and peptidase inhibitor classification is based on MEROPS nomenclature and is detailed in Appendix A. Life cycle stage abbreviations: Mir, miracidia; Red, rediae; Cer, cercariae; Met, metacercariae; NEJ1hr, NEJ 1hr post-excystment; NEJ3hr, NEJ 3hr post-excystment; NEJ24hr, NEJ 24hr post-excystment; Immature, immature flukes 21 days post infection (dpi); Juv_42d, immature flukes 42 dpi; Juv_70d, immature flukes 70 dpi. The graphs were generated by ggplot2 in R.

**Figure 3 genes-13-01854-f003:**
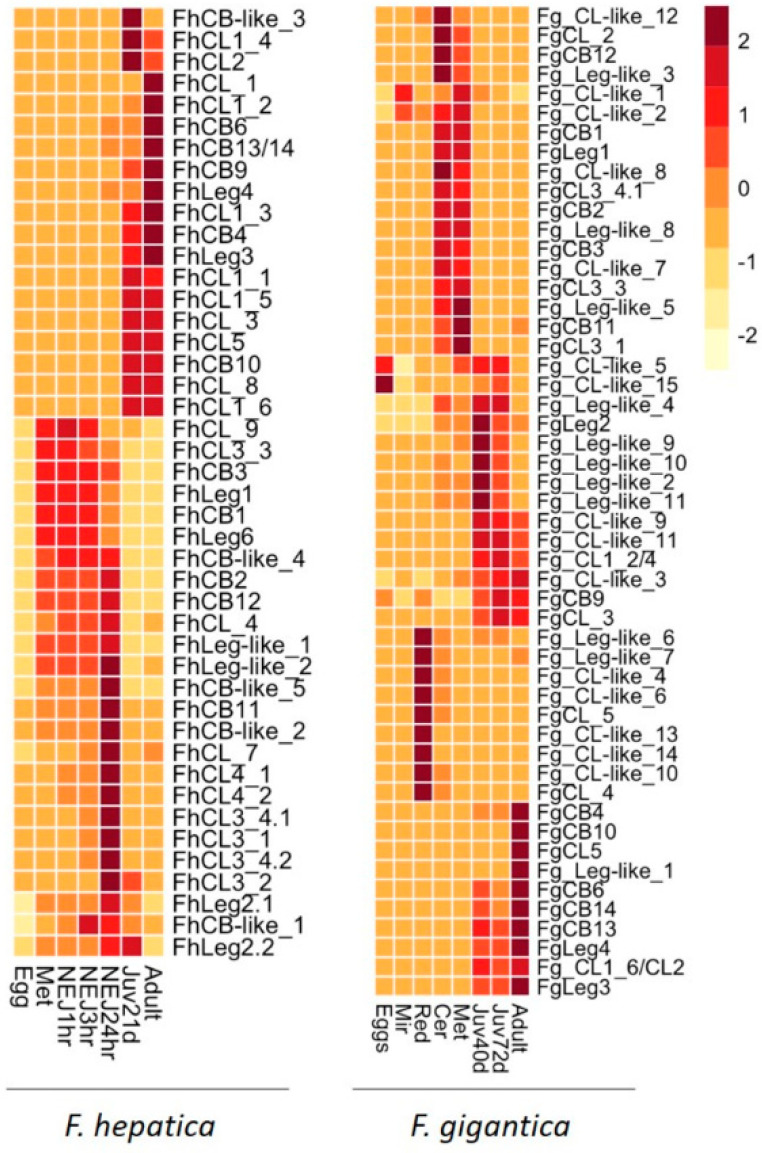
Differential gene expression of the cathepsin L and B peptidase and the legumain genes. Genes expressed by biological replicates of the respective life cycle stages from the *F. hepatica* and *F. gigantica* stage-specific transcriptome datasets were grouped by hierarchical clustering, represented by a heatmap generated by pheatmap in R. Up-regulation represented in dark red; down-regulation represented in light yellow. The annotation of the *F. hepatica* genes is based on the analysis by Cwiklinski et al. [4] and the annotation of the *F. gigantica* genes is based on the comparative analysis described in this study (detailed in Appendix A).

**Figure 4 genes-13-01854-f004:**
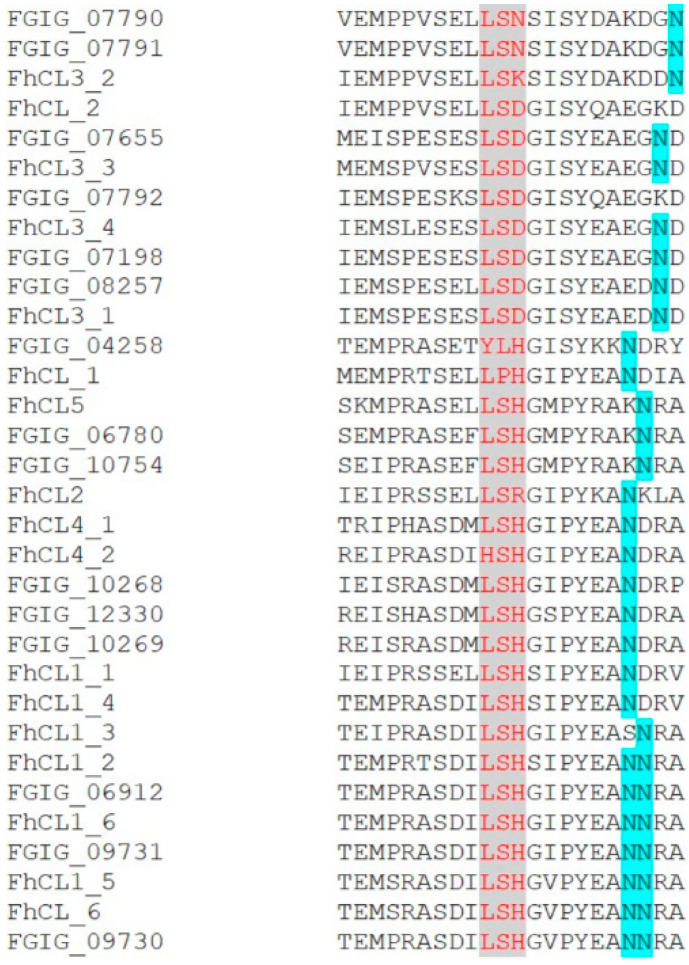
Alignment of the amino acid sequence spanning the junction between the propeptide and mature domain of the *F. hepatica* and *F. gigantica* cathepsin L peptidase families. The conserved cathepsin L intra-molecular cleavage site and the conserved asparagine (N) of the legumain/asparaginyl endopeptidase trans-molecular cleavage site are highlighted in grey/red and blue, respectively. The *F. hepatica* cathepsin L peptidase classification is based on the study by Cwiklinski et al. [4]. The *F. gigantica* sequences are from the *F. gigantica* genome (PRJNA230515).

**Figure 5 genes-13-01854-f005:**
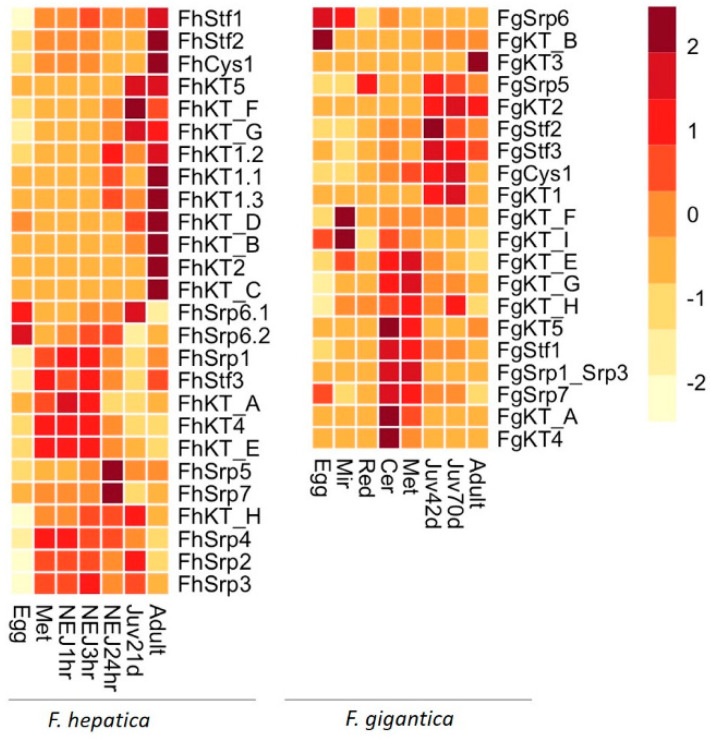
Differential gene expression of the cathepsin cysteine peptidase and serine peptidase inhibitor genes. Genes representing the Kunitz-type inhibitors, serpins and cystatins/stefins, expressed by biological replicates of the respective life cycle stages from the *F. hepatica* and *F. gigantica* stage-specific transcriptome datasets, were grouped by hierarchical clustering, represented by a heatmap generated by pheatmap in R. Up-regulation represented in dark red; down-regulation represented in light yellow. The annotation of the *F. hepatica* genes is based on the analysis described herein and by [5,7,9], and the annotation of the *F. gigantica* genes is based on the comparative analysis described in this study (detailed in Appendix A).

**Figure 6 genes-13-01854-f006:**
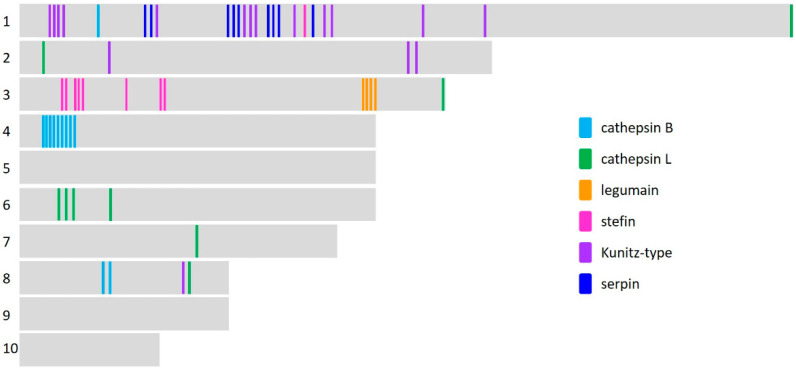
Graphical representation of the chromosomal location of the cathepsin peptidase, legumain and peptidase inhibitors genes within the *F. gigantica* genome. The schematic is not drawn to scale but displays the relative position of the genes within the ten chromosomes.

## Data Availability

The genome and transcriptome datasets interrogated as part of this study are available at WormBase ParaSite (https://parasite.wormbase.org; accessed on 21 July 2022) and in the public repositories as follows: (a) *F. hepatica* genome data reported by Cwiklinski et al. [12] available at WormBase ParaSite and NCBI/ENA: PRJEB6687 and PRJEB25283; (b) *F. hepatica* stage-specific transcriptome data reported by Cwiklinski et al. [12] available at WormBase ParaSite and the NCBI/ENA: PRJEB6904 and Ilgová et al. [30] available at NCBI Gene Expression Omnibus: GSE160622; (c) *F. gigantica* genome data reported by Choi et al. [14], available at WormBase ParaSite and NCBI/ENA: PRJNA230515 and Luo et al. [15] available at NCBI: PRJNA691688 and Genome Warehouse: GWHAZTT00000000 (d) *F. gigantica* stage-specific transcriptome data reported by Zhang et al. [26] available at NCBI/ENA: PRJNA350370. The mass spectrometry proteomics data analysed as part of this study have been deposited to the ProteomeXchange Consortium via the PRIDE partner repository with the following data set identifiers (a) egg datasets (Ilgová et al. [30]): PXD022516; (b) metacercariae and NEJ specific datasets (Cwiklinski et al. [27]): PXD007255, PXD016561; (c) immature fluke (Cwiklinski et al. [28]): PXD021221; (d) adult ES and EV datasets (Murphy et al. [29]): PXD002570 and PXD016561.

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
