# Peer review of "Exploiting Comparative Omics to Understand the Pathogenic and Virulence-Associated Protease: Anti-Protease Relationships in the Zoonotic Parasites Fasciola hepatica and Fasciola gigantica"

_genes, 2022, doi:10.3390/genes13101854_

Round 1

Reviewer 1 Report

Introduction

The final paragraph must state very clearly the objectives of the work

Materials and methods

It is not clear if the authors used data from own isolates or whether they only employed previous findings that exist in databases.

This must be mentioned very clearly and explicitly.

Results

The lack of clear definition of the source of information becomes evident in this section.

Discussion

Please add a new subsection with the clinical implications of these findings.

Reviewer 2 Report

The authors compared the sequences of peptidase and peptidase inhibitors in F. gigantica and F. hepatica genomes and found several differences between these two closely related species. The approach is quite original, and their findings provide interesting and promising leads for further investigations.

The authors are one of the leading international experts on Fasciola biology. More than half of the references cited were authored by one, or both, of the authors.

The paper is very well written, and data, including tables and figures, are of excellent quality.

MAJOR COMMENTS:

None

MINOR COMMENTS:

Line 12 (Abstract): “(worm)” can be deleted.

Lines 26 (Abstract) and 444-445, “Fasciola spp. hybrids that are spreading throughout Asia;” “In areas where F. hepatica and F. gigantica overlap, such as in Southeast Asia, China and Korea, species hybridization has been observed”: Recent studies conducted elsewhere seem to suggest that these two Fasciola species may be sympatric in other continents (e.g. in Africa and also in the Middle East) and that hybrids may occur outside Asia. Please see some of the references cited below. Any comments?

-Omar MA, Elmajdoub LO, Ali AO, Ibrahim DA, Sorour SS, Al-Wabel MA, Ahmed AI, Suresh M, Metwally AM. Genetic characterization and phylogenetic analysis of Fasciola species based on ITS2 gene sequence, with first molecular evidence of intermediate Fasciola from water buffaloes in Aswan, Egypt. Ann Parasitol. 2021;67(1):55-65. doi: 10.17420/ap6701.312.

-Haridwal S, Malatji MP, Mukaratirwa S. Morphological and molecular characterization of Fasciola hepatica and Fasciola gigantica phenotypes from co-endemic localities in Mpumalanga and KwaZulu-Natal provinces of South Africa. Food Waterborne Parasitol. 2021;22:e00114. doi: 10.1016/j.fawpar.2021.e00114.

-Giovanoli Evack J, Schmidt RS, Boltryk SD, Voss TS, Batil AA, Ngandolo BN, Greter H, Utzinger J, Zinsstag J, Balmer O. Molecular confirmation of a Fasciola gigantica × Fasciola hepatica hybrid in a Chadian Bovine. J Parasitol. 2020;106(2):316-322. doi: 10.1645/19-66.

-Mucheka VT, Lamb JM, Pfukenyi DM, Mukaratirwa S. DNA sequence analyses reveal co-occurrence of novel haplotypes of Fasciola gigantica with F. hepatica in South Africa and Zimbabwe. Vet Parasitol. 2015;214(1-2):144-51. doi: 10.1016/j.vetpar.2015.09.024.

-Ashrafi K, Valero MA, Panova M, Periago MV, Massoud J, Mas-Coma S. Phenotypic analysis of adults of Fasciola hepatica, Fasciola gigantica and intermediate forms from the endemic region of Gilan, Iran. Parasitol Int. 2006;55(4):249-60. doi: 10.1016/j.parint.2006.06.003.

Line 179: …proteome reveals that high levels of cysteine peptidases are expressed, OR …proteome reveals high levels of cysteine peptidases that are expressed

Lines 349-351, “stefins mapped to 8 genes all located on chromosome 3, whereas the multi-domain cystatin located to chromosome 1”: In Fig. 6, there seem to be 9 (rather than 8) stefin genes.

Lines 352-354, Kunitz-type inhibitors “The single domain Kunitz-type inhibitors are all located on chromosome 1 with the exception of FgKT4 which is located on chromosome 4”: In Fig. 6, FgKT4 does not seem to be represented on chromosome 4.

Lines 363-368, “…at higher temperature thresholds for the F. gigantica intra-snail stages…have been reported” Please check the syntax of this part of the sentence.

Lines 372-374, “Advances in sequencing technologies and the continual reduction in their costs [delete the comma here; the subject of the sentence is plural] have facilitated… and provide (or have provided)…”

Line 382, “parasitology-related functions”: parasite-related functions?

Lines 409-410, “an abundance…are observed”: The word “abundance” is a singular noun.

Line 422: Delete the comma after “function”

References: Please use the same format throughout. See for example the article title of REF 6 and 50 (the first letters of some words are in capital letter).

REF 1: Is there something missing or wrong with the name of the last author? Please check.

REF 7: Vaccines (Basel) 2022, 10, 155

REF 12: Genome Biol 2015, 16, 71.

REF 18, 19: Acta Trop

REF 27: D624-D632

REF 51: Parasite Immunol

Round 2

Reviewer 1 Report

The authors should improve the legends of the figures, making them more concise, before final acceptance.

Author Response

Reviewer's comment: The authors should improve the legends of the figures, making them more concise, before final acceptance.

Our response: We have included all the relevant information in the figure legends in accordance with the journal's guidelines. Making the figure legends more concise would remove vital information required for the interpretation of the figures; as such we believe the legends should remain as there are.